Probiotic potential of bacteria associated with the mangrove epiphytic algae Bostrychia calliptera and Rhizoclonium riparium

Martinez-Delgado Juliana martinez.juliana@correounivalle.edu.co
Benitez-Campo Neyla
Department of Biology, Universidad del Valle , Cali , Valle del Cauca , Colombia
Esteban María Ángeles
Electronic publication date: 2025 Jun 3
Publication date: 2025
Volume: 13
Electronic Location ID: e19073
Received 2024 Nov 7; Accepted 2025 Feb 10
Copyright: ©2025 Martinez-Delgado and Benitez-Campo
Copyright year: 2025
Copyright holder: Martinez-Delgado and Benitez-Campo
License: This is an open access article distributed under the terms of the Creative Commons Attribution License, which permits unrestricted use, distribution, reproduction and adaptation in any medium and for any purpose provided that it is properly attributed. For attribution, the original author(s), title, publication source (PeerJ) and either DOI or URL of the article must be cited.
License URL: https://creativecommons.org/licenses/by/4.0/

Keywords: Avicennia germinans, Rhizophora mangle, Culturable diversity, Biofilms, Antibiotic susceptibility, Tolerance tests

Funding: The Fondo Ciencia, Tecnología e Innovación del Sistema General de Regalías BPIN: 2021000100488 The Universidad del Valle CI:71323 This research was funded by the Fondo Ciencia, Tecnología e Innovación del Sistema General de Regalías (BPIN: 2021000100488) and by the Universidad del Valle (CI:71323). The funders had no role in study design, data collection and analysis, decision to publish, or preparation of the manuscript.

==============================
The growth of the global population has driven the development of aquaculture as an alternative means of meeting the increasing demand for food. However, this sector faces challenges from pathogen transmission, which can impact both cultured organisms and consumers. Probiotics offer a promising solution by leveraging the antibacterial activity of certain microorganisms against pathogens. Given the limited research on the probiotic potential of microorganisms associated with marine algae, this study aimed to evaluate this potential of bacteria isolated from Bostrychia calliptera and Rhizoclonium riparium, algae from mangroves on the Pacific coast of Colombia. The antibacterial activity of the isolates was evaluated against six fish and shellfish pathogens, leading to the selection of four strains (Bacillus sp. AB08, Bacillus sp. AB17, Bacillus sp. AN35, and Pseudomonas mosselii AR37) as probiotic candidates due to their outstanding inhibition of Staphylococcus aureus. None of the selected strains formed biofilms, a favorable result from a pathogenicity perspective, while Bacillus sp. AB08 and AN35 demonstrated notable susceptibility to all tested antibiotics. Additionally, these two strains exhibited broad tolerance to temperature and pH, maintaining viable counts above 106 CFU/mL, characteristics that position them as promising candidates for use as probiotics. However, further in vitro studies are needed to better define their probiotic properties, along with in vivo evaluations in aquaculture systems to confirm their efficacy and safety.

Introduction

Aquaculture has become a crucial component in addressing the growing global demand for food, offering products that play a significant role in the human diet (Wang, Li & Li, 2015; Abdel-Latif et al., 2022; Fazle-Rohani et al., 2022). Today, the majority of seafood and fish consumed worldwide originate from aquaculture farms, whose production has steadily increased over recent decades (Wang, Li & Li, 2015; Khouadja et al., 2017; FAO, 2024). However, this intensive production practice increases the susceptibility of cultured aquatic species to diseases, leading to developmental damage, high mortality rates, challenges in controlling outbreaks, and substantial economic losses (Fazle-Rohani et al., 2022; Mujeeb et al., 2022). Additionally, contaminated aquaculture products can act as vectors for pathogenic microorganisms, posing severe health risks to humans, including illnesses such as listeriosis, botulism, cholera, and gastrointestinal infections which may result in dehydration, inflammatory responses, and even death in severe cases (Feldhusen, 2000; Teplitski, Wright & Lorca, 2009; Wang, Li & Li, 2015; Elbashir et al., 2018; Ali et al., 2020; Mendes et al., 2023). Common pathogens associated with fish and shellfish include Escherichia coli, Klebsiella spp., Clostridium botulinum, Listeria monocytogenes, Salmonella spp., Staphylococcus aureus, Aeromonas hydrophila, and Vibrio spp. (Ghaderpour et al., 2014; Wang, Li & Li, 2015; Poharkar et al., 2016).

Given these problems, the asepsis of aquaculture food products, particularly those traditionally consumed raw, has become a growing priority (Teplitski, Wright & Lorca, 2009; Mendes et al., 2023). Conventionally, pathogen management in aquaculture has relied on antibiotics, sterilization agents, prophylactics, and chemotherapeutics (Fazle-Rohani et al., 2022; Mujeeb et al., 2022). However, excessive use of these measures has resulted in antibiotic-resistant pathogens and environmental contamination caused by the accumulation of harmful chemical residues (Butkhot et al., 2020; Abdel-Latif et al., 2022; Fazle-Rohani et al., 2022; Mujeeb et al., 2022). As a result, the demand for cost-effective, efficient, environmentally safe, and non-invasive alternatives to control fish and shellfish pathogens has grown. In this context, probiotics have emerged as a promising and sustainable solution (Khouadja et al., 2017; Butkhot et al., 2020; Abdel-Latif et al., 2022).

Probiotics, defined as live microorganisms that provide health benefits to the host when administered in adequate amounts (Verschuere et al., 2000; Bidhan et al., 2014; Abdel-Latif et al., 2022), offer significant advantages for aquaculture. They enhance feed efficiency by producing digestive enzymes, thereby increasing the growth rates and nutritional value of aquatic organisms. Probiotics also play a crucial role in maintaining healthy aquatic ecosystems by improving water quality and shaping the bacterial composition of water and sediments. Additionally, probiotics strengthen the host’s immune response by stimulating immunity, competing with pathogens for resources, and producing antimicrobial compounds (Verschuere et al., 2000; Irianto & Austin, 2002; Vieira et al., 2013; Bidhan et al., 2014; Mujeeb et al., 2022). However, for probiotics to provide these benefits, they must meet safety criteria, including antibiotic susceptibility and the absence of pathogenic traits, such as biofilm formation, which can increase antibiotic resistance and facilitate tissue colonization (Sanders et al., 2010; Reichling, 2020; Cheong et al., 2021; Mujeeb et al., 2022). Furthermore, for probiotics to be effective, they must meet functional properties such as withstanding the harsh conditions of host target organs and production processes, such as low pH and high temperatures, while maintaining viable counts above 106 CFU/ml, the minimum recommended threshold to ensure probiotic efficacy (Tripathi & Giri, 2014; Jiang et al., 2018; Pang, Ransangan & Hatai, 2020).

Mangrove ecosystems host diverse microbial communities adapted to fluctuating conditions, including salinity, flooding, light, and temperature (Bouchez et al., 2013; Hwanhlem, Chobert & H-Kittikun, 2014). Many of these microorganisms produce bioactive compounds with applications in nutraceutical, pharmaceutical, agrochemical, and food industries (Rishad et al., 2016; Chukwudulue et al., 2023; Pereira et al., 2023). Among these, epiphytic bacteria play a crucial ecological role in protecting algae, which lack an immune system, by producing antimicrobial compounds that defend them against pathogens (Busetti, Maggs & Gilmore, 2017; De Mesquita et al., 2019; Chukwudulue et al., 2023). Due to their ecological functions and antimicrobial potential, these bacteria have emerged as a promising source of probiotics for aquaculture.

Despite their potential, little is known about which epiphytic bacteria from mangrove-associated algae possess probiotic properties or the ability to produce bioactive compounds for aquaculture and related industries. This knowledge gap is particularly evident in Colombia, despite the country’s extensive mangrove ecosystems along the Pacific coast. These mangrove forests support epiphytic algae such as Bostrychia calliptera (Rhodophyta) and Rhizoclonium riparium (Chlorophyta) (Peña-Salamanca, 2008; Rengifo-Gallego, Peña-Salamanca & Benitez-Campo, 2012; Thatoi et al., 2013; Cantera & Londoño, 2017). These algae are commonly associated with the roots of Rhizophora mangle (red mangrove) and the pneumatophores of Avicennia germinans (black mangrove), the dominant mangrove species in the region (Peña-Salamanca, 2008). To date, no studies have explored the bacterial communities associated with these algae or their potential as probiotics. Therefore, this study aims to address this gap by characterizing epiphytic bacteria from mangrove algae, B. calliptera and R. riparium evaluating their probiotic potential based on antimicrobial activity, safety aspects such as biofilm formation and antibiotic susceptibility, and their ability to adapt to aquaculture-relevant conditions.

Materials and Methods

Sampling

Samples were collected from a mangrove forest in the Dagua river delta, Valle del Cauca, Colombia, with two sampling stations established (station 1: 3°51′25.9″N, 77°04′16.9″W and station 2: 3°51′5.161″N, 77°3′39.409″W) (Fig. 1). Three R. mangle and three A. germinans trees, each hosting roots and pneumatophores colonized by the algae B. calliptera and R. riparium, were selected for bacterial isolation. Algal surfaces were swabbed with sterile cotton swabs and transferred to Falcon tubes containing synthetic seawater (SSW) (Nguyen, 2018). Samples were stored at 4 °C and transported to the Microbiological Research Laboratory at Universidad del Valle for further processing. Additionally, specimens of B. calliptera were collected, stored at 4 °C, and maintained in an aquarium with F/2 medium (Lananan et al., 2013) at a salinity of 20 ppm.

Figure 1 Map of the sampling sites in the mangrove area of Buenaventura, Colombia.

Map data ©OpenStreetMap contributors. Data available under the Open Database License (https://www.openstreetmap.org/copyright).

The Universidad del Valle holds a Collection Framework Permissions (Resolution 1070, August 28, 2015) issued by the Autoridad Nacional de Licencias Ambientales (ANLA) of the Ministerio de Ambiente y Desarrollo Sostenible. This permit authorizes academic programs, research groups, and faculty members to collect wild biological specimens for non-commercial scientific research purposes.

Isolation and morphological characterization of bacterial strains associated with B. calliptera and R. riparium

To isolate bacterial strains associated with B. calliptera and R. riparium, 10 mL of the combined replicates from each smear were inoculated into 90 mL of SSW, soy flour mannitol medium (SFM) (Hobbs et al., 1989), and tryptic soy broth (TSB) at 28 °C with shaking at 180 rpm. The SFM and TSB cultures were incubated for 24 h, while the SSW cultures were incubated for 21 days. After incubation, 100 µL aliquots were plated on the respective media using the standard plate count method (Madigan et al., 2015). Additional isolates were obtained by swabbing the surface of B. calliptera maintained in an aquarium and inoculating the swabs onto SFM, trypticase soy agar (TSA), SSW, and nutrient agar (NA), followed by incubation at 28 °C for 24 to 48 h.

Macroscopic observation of colony growth was performed, and colonies exhibiting distinctive morphological features were designated as unique strains, resulting in an initial selection of 80 strains. These were subsequently subcultured in NA, TSA, SFM and SSW, until uniform colonies were obtained. Gram staining and microscopic characterization were performed to confirm purity and assess cell shape, size, and arrangement. Strains displaying identical morphological features or signs of contamination (evidenced by two distinct morphologies at the microscopic level) were excluded, yielding a final collection of 56 bacterial strains. Finally, all bacterial isolates were stored at −70 °C for further analysis.

Biocontrol testing against fish and shellfish pathogens

The biocontrol potential of 56 bacterial strains isolated from B. calliptera and R. riparium was evaluated against S. aureus (ATCC 29737), E. coli (ATCC 11229), L. monocytogenes (ATCC 13932), Salmonella bongori (ATCC 43975), Vibrio brasiliensis (RR81) and A. hydrophila (RB65), all of which are recognized fish and shellfish pathogens (Ghaderpour et al., 2014; Wang, Li & Li, 2015; Poharkar et al., 2016).

The agar diffusion method (Bauer et al., 1966) was employed with modifications. Bacterial inocula were prepared in TSB, LB broth, or SSW, depending on the specific requirements of each strain, and incubated at 28 °C for 24 h. Subsequently, the inocula concentration was adjusted to an optical density (OD) of 0.08–0.1 at 625 nm, equivalent to the 0.5 McFarland standard (European Committee on Antimicrobial Susceptibility Testing (EUCAST), 2024).

Pathogens were then spread onto Müller-Hinton agar (MHA) plates using sterile swabs, and bacterial isolates were transferred to the center of the pre-inoculated MHA plates using a 96-pin microplate replicator. As positive and negative controls, 30 µg chloramphenicol disks and sterile saline solution were used, respectively. All assays were performed in triplicate at 28 °C, the optimal breeding temperature for tropical fish and shellfish (Boyd, 2018), and at 37 °C, corresponding to human body temperature (Cramer et al., 2022). The incubation period was 18 h.

Strains exhibiting inhibition zones were further evaluated using the disk diffusion method, as described by Benítez-Campo, Vivas Zarate & Rosero Hernandez (2009) and Huang et al. (2023), with modifications. Sterile Whatman filter paper disks (six mm in diameter) were impregnated with 10 µL of the selected bacterial inoculum and placed on MHA plates pre-inoculated with pathogens. Finally, inhibition zone radii were measured from the edge of the filter paper disk to the boundary of the inhibition area. Inhibition levels were categorized according to the modified criteria of Wanja et al. (2020): no inhibition (0 mm), incipient inhibition (1–5 mm), moderate inhibition (6–9 mm), and strong inhibition (>10 mm). Strains exhibiting the highest biocontrol activity were selected for further evaluation.

Molecular identification of selected bacterial strains

Bacterial isolates were cultured in Eppendorf tubes containing one mL of TSB and incubated at 28 °C for 24–48 h. After incubation, samples were centrifuged, washed twice with phosphate-buffered saline (PBS), and resuspended in 100 µL of PBS. DNA extraction was performed using the Monarch DNA extraction kit (New England Biolabs, Ipswich, MA, USA) according to the manufacturer’s instructions.

The 16S rRNA gene was amplified using the universal primers 63F and 1387R (Marchesi et al., 1998). The PCR master mix contained 2.5 µL of 10X buffer, 0.5 µL of dNTPs, 0.5 µL of each primer, 0.125 µL of Taq polymerase, 1 µL of DNA, and ultrapure water to a final volume of 25 µL. PCR conditions included an initial denaturation at 95 °C for 30 s, followed by 30 cycles of denaturation at 95 °C for 30 s, annealing at 55 °C for 1 min, extension at 68 °C for 1:20 min, and a final extension at 68 °C for 10 min. PCR products were sequenced by Macrogen (Korea), and sequences were analyzed using the Basic Local Alignment Search Tool (BLAST) from the National Center for Biotechnology Information (NCBI) to identify bacterial isolates.

Phylogenetic analysis

The phylogenetic relationships of bacterial strains selected from the biocontrol tests were inferred using partial 16S rRNA gene sequences. These sequences were compared to closely related bacterial species identified via BLAST and to representative sequences from the corresponding genera in the NCBI database. Streptomyces griseus strain KACC 20084 (NR_042791.1) and Streptomyces nigrescens strain NRRL ISP-5276 (NR_116013.1) were included as outgroups to root the phylogenetic tree.

Sequence alignment was performed using the Muscle algorithm (Edgar, 2004a; Edgar, 2004b), and phylogenetic reconstruction was conducted via Bayesian inference in BEAST2 (v2.6.7). The general time reversible (GTR) substitution model (Tavaré, 1986) was applied, and the robustness of the tree topology was assessed through a Markov Chain Monte Carlo (MCMC) simulation run for 10 million generations to ensure convergence.

Assessment of the probiotic safety of bacterial isolates associated with algae

Biofilm formation assay

The biofilm-forming ability of the selected bacterial strains was evaluated using a modified crystal violet (CV) assay (Stepanović et al., 2007). Standardized bacterial inocula (OD600 = 0.1–0.5) were added in quadruplicate to a 96-well microplate containing TSB and incubated at 28 °C for 48 h. Control wells included TSB (blank), E. coli (ATCC 11229, negative control), and Pseudomonas aeruginosa (ATCC 27853, positive control) (El-Abed et al., 2011). After incubation, wells were washed with PBS (pH 7.2) at room temperature, air-dried, and heat-fixed at 60 °C for 1 h. Biofilms were stained with CV for 10 min, followed by resolubilization with 95% ethanol for 30 min. Absorbance at 570 nm was measured using a microplate reader. Biofilm production was classified according to Stepanović et al. (2007) into non-producers, weak, moderate, and strong producers.

Antibiotic susceptibility test

The antibiotic susceptibility of bacterial strains was assessed using the disk diffusion method on MHA (Bauer et al., 1966). Bacterial cultures were standardized to an OD625 of 0.08–0.1 and spread onto the surface of MHA plates. The antibiotics tested included streptomycin (10 µg), ciprofloxacin (5 µg), tetracycline (30 µg), oxytetracycline (30 µg), ampicillin (20 µg), chloramphenicol (30 µg), kanamycin (30 µg), and penicillin G (10 IU). Plates were incubated at 28 °C for 18 h. The radius of the inhibition zones was measured and the results were classified according to the guidelines of Patel et al. (2009) and Ramesh et al. (2015), with modifications: resistant (≤5 mm), sensitive (6–9 mm) and highly sensitive (≥10 mm).

Evaluation of probiotic functional properties of bacterial isolates associated with algae

Tolerance testing for temperature and pH

Temperature tolerance was assessed by inoculating the selected strains into tubes containing two mL of TSB and incubating for 16 h at temperatures of 25, 28, 37, 45, 50, 55, and 60 °C. Colony forming units (CFU/mL) were quantified using the massive stamping plate drop method (Corral-Lugo et al., 2012), with 10 µL aliquots of dilutions (10−3 to 10−7) placed on TSA plates, followed by 16 h of incubation at the same temperatures.

For pH tolerance, the strains were inoculated into TSB adjusted to pH levels ranging from two to nine. Cultures were incubated at 28 °C for 16 h, and CFU/mL was determined using the same massive stamping plate drop method, with incubation at 28 °C for 16 h.

Statistical analysis

The capacity of the temperature and pH tolerances was subjected to statistical analysis using the RStudio software (version 4.3.3). Data were log-transformed (base 10) and analyzed using ANOVA to determine statistical significance (p-value < 0.05).

Results

Biocontrol of fish and shellfish pathogens

Twelve of the 56 bacterial isolates tested (21.42%) inhibited the growth of at least one of six fish and shellfish pathogens (Table 1). It is noteworthy that S. aureus was the most susceptible pathogen, as it was inhibited by all twelve strains, with inhibition levels ranging from incipient (II) to moderate (IM) at both temperatures. Furthermore, none of the strains tested inhibited the growth of the other pathogens tested: V. brasiliensis, S. bongori, L. monocytogenes or E. coli.

Several strains, including AB01, AB07, AB09, and AR29, exhibited consistent inhibition (incipient, II) at both temperatures against S. aureus and A. hydrophila. Strains AB08, AB17, AN35, and AR37 stood out for exhibiting moderate inhibition (MI) against S. aureus, although only at 28 °C. In the case of A. hydrophila, half of the strains tested exhibited inhibitory activity, reaching incipient inhibition (II) at both temperatures. However, certain strains, including AR20, AR28, AR31, AR37, and AN35, showed no inhibition (NI) against this pathogen under any of the conditions tested.

In consideration of these observations, the antimicrobial activity of strains AB08, AB17, AR37 and AN35 against S. aureus (Fig. 2) led to their selection for further testing to evaluate their potential as probiotics for aquaculture.

Table 1 Inhibitory behavior of 12 bacterial isolates against two fish and shellfish pathogens at 28 °C and 37 °C.

Strain	S. aureus	A. hydrophila	
	28 °C	37 °C	28 °C	37 °C	
AB01	II	II	II	II	
AB02	II	II	NI	II	
AB07	II	II	II	II	
AB08	MI	II	II	II	
AB09	II	II	II	II	
AB17	MI	II	II	II	
AR20	II	II	NI	NI	
AR28	II	II	NI	NI	
AR29	II	II	II	II	
AR31	II	II	NI	NI	
AR37	MI	II	NI	NI	
AN35	MI	II	NI	NI	
C+	SI	SI	SI	SI	
Notes.

C+: Positive control (Chloramphenicol, 30 μg).

No inhibition (NI): 0 mm, Incipient inhibition (II): 1–5 mm, Moderate inhibition (MI): 6–9 mm, Strong inhibition (SI): >10 mm.

Figure 2 Inhibition of S. aureus by the disk diffusion method of bacteria associated with selected algae (AB08, AB17, AR37 and AN35).

(A) At 28 °C. (B) At 37 °C. C+ indicates the positive control (chloramphenicol, 30 µg) and C- indicates the negative control.

Morphological characterization and molecular identification of the selected bacterial strains

Strains AB08, AB17, and AN35 were Gram-positive, rod-shaped bacteria, with cell sizes ranging from 1.96 to 2.45 µm (Figs. 3A, 3B and 3C). On AN, these strains formed colonies with similar characteristics: cream-colored, circular, mucous, and slightly opaque, with entire borders and flat elevations (Figs. 3E, 3F and 3G). In contrast, AR37, a Gram-negative rod-shaped bacterium of approximately 0.98 µm in size (Fig. 3D), formed small, opaque, circular, cream-colored colonies with entire edges and flat elevations on NA (Fig. 3H), accompanied by a slightly yellowish coloration in the medium.

Figure 3 Microscopic (A–D) and macroscopic (E–H) morphological characteristics of the four selected strains.

Microscopic images (100x) correspond to: (A) Bacillus sp. AB08. (B) Bacillus sp. AB17. (C) Bacillus sp. AN35. (D) P. mosselii AR37. Macroscopic images show colonies on nutrient agar of: (E) Bacillus sp. AB08. (F) Bacillus sp. AB17. (G) Bacillus sp. AN35. (H) P. mosselii AR37.

BLAST analysis revealed that three of the four strains selected in the biocontrol tests (AB08, AB17 and AN35) were associated with the genus Bacillus, with identical identity percentages across multiple species within this group. Meanwhile, strain AR37 exhibited 99.81% similarity to Pseudomonas mosselii, exceeding the species threshold value (98.7%) (Ismail et al., 2018).

The phylogenetic tree (Fig. 4) showed that strains AB08, AN35 and AB17 formed a well-supported clade with members of Bacillus, exhibiting a posterior probability value of 100%. However, AN35 and AB17 did not cluster with any specific species, while AB08 was related to Bacillus stercoris D7XPN1 (NR_181952.1) but with a very low support value (22.23%), thus precluding precise species identification of these three isolates. On the other hand, strain AR37 was found to be closely related to the sequence of P. mosselii CFML 90-83 (NR_024924.1) with a posterior probability of 100%. This indicates a high probability of belonging to this species and confirms the molecular identification that was performed using BLAST. Based on these results, the four selected strains were identified as Bacillus sp. AB08 (PQ276483), Bacillus sp. AB17 (PQ276479), Bacillus sp. AN35 (PQ276436) and P. mosselii AR37 (PQ276466).

Figure 4 Phylogenetic tree of the relationship between the four selected strains with the highest biocontrol activity (in red) and their phylogenetically closest species.

Numbers and circles at the nodes indicate posterior probability values (in percent) calculated from 10 million generations.

Probiotic safety

Biofilm formation

The biofilm formation assay revealed that none of the selected strains were capable of forming biofilms, a trait considered beneficial for the safety of certain probiotics.

Susceptibility to antibiotics

The antibiotic susceptibility tests revealed that the strains of the Bacillus genus exhibited the highest overall susceptibility. Notably, both Bacillus sp. AB08 and Bacillus sp. AN35 demonstrated sensitivity to all tested antibiotics, displaying an identical sensitivity pattern. However, Bacillus sp. AB17 exhibited resistance to streptomycin. In contrast, P. mosselii AR37 exhibited the highest degree of antibiotic resistance, demonstrating sensitivity only to ciprofloxacin and kanamycin (Table 2).

Table 2 Antibiotic susceptibility profile of the four selected bacterial strains.

Strain	Antibiotics	
	Aminoglycosides	Fluoroquinolones	Tetracyclines	Penicillins	Amphenicols	
	S	K	Cip	TE	OT	SAM	P	C	
Bacillus sp. AN35	S	HS	HS	HS	S	HS	HS	HS	
Bacillus sp. AB08	S	HS	HS	HS	S	HS	HS	HS	
Bacillus sp. AB17	R	HS	HS	HS	S	HS	HS	HS	
P. mosselii AR37	R	S	HS	R	R	R	R	R	
Notes.

S Streptomycin

K Kanamycin

Cip Ciprofloxacin

TE Tetracycline

OT Oxytetracycline

SAM Ampicillin

C Chloramphenicol

P Penicillin G

Resistant (R): ≤5 mm, Sensitive (S): 6–9 mm, Highly sensitive (HS): ≥ 10 mm.

Probiotic functional properties

Temperature tolerance

The Bacillus strains (AB08, AB17, and AN35) exhibited a high growth capacity within a temperature range of 25 to 55 °C, maintaining counts equal to or greater than 107 CFU/mL throughout this interval. However, their growth was completely inhibited at 60 °C. In contrast, P. mosselii AR37 displayed a more restricted tolerance range, achieving counts exceeding 108 CFU/mL between 25 and 37 °C but experiencing total growth inhibition at 45 °C. Statistical analysis revealed significant differences among the strains in their response to the evaluated temperatures (p < 0.0001) (Fig. 5).

Figure 5 Growth of selected strains exposed to different temperatures.

Error bars represent standard deviation (±SD). Superscripts indicate statistically significant differences (p < 0.05) within each strain. The red line shows the minimum recommended count (106 CFU/mL) for effective probiotics.

pH tolerance

The Bacillus strains (AB08, AB17, and AN35) demonstrated broad pH tolerance, maintaining counts equal to or greater than 106 CFU/mL across all evaluated pH levels (two to nine), except for Bacillus sp. AB17, which exhibited a count of 105 CFU/mL at pH 2 (p < 0.0001), a value considered below the minimum threshold recommended for an effective probiotic. In contrast, P. mosselii AR37 exhibited growth exceeding 107 CFU/mL within a narrower pH range (five to nine) but failed to survive at lower pH values. Statistical analysis confirmed significant differences among the strains in their response to the various pH conditions (p < 0.0001) (Fig. 6).

Figure 6 Growth of selected strains exposed to different pH levels.

Error bars represent standard deviation (±SD). Superscripts indicate statistically significant differences (p < 0.05) within each strain. The red line shows the minimum recommended count (106 CFU/mL) for effective probiotics.

Discussion

Marine ecosystems, including mangrove forests, are renowned for their microbial diversity, encompassing free-living microorganisms and those attached to natural or artificial surfaces (Mieszkin, Callow & Callow, 2013; Ravisankar, Gnanambal & Sundaram, 2013; Kaur et al., 2023). Bacteria associated with algae have a remarkable capacity to produce antimicrobial compounds, with studies reporting that 35–50% of bacterial isolates from algae exhibit antimicrobial activity (Goecke et al., 2010; Thatoi et al., 2013; Albakosh et al., 2016; Ismail et al., 2018; De Mesquita et al., 2019). In this study, a slightly lower antimicrobial activity (21.42%) was observed, probably due to the culture media used during isolation, which may have limited the diversity of the recovered bacteria. Nevertheless, the isolates evaluated exhibited notable biocontrol activity against S. aureus and A. hydrophila. Among these, Bacillus sp. AN35, Bacillus sp. AB08, Bacillus sp. AB17, and P. mosselii AR37 demonstrated the highest effectiveness, showing moderate inhibition zones (6–9 mm) at 28 °C against S. aureus (Table 1). These results highlight the potential of algal-associated bacteria as a source of novel antimicrobial compounds with biotechnological applications. Furthermore, the enhanced activity of these strains at 28 °C, an optimal temperature for aquaculture in tropical regions (Boyd, 2018), positions them as promising candidates for aquaculture use in tropical zones.

Some studies have demonstrated the antimicrobial efficacy of probiotic Bacillus and Pseudomonas species in controlling fish and shellfish pathogens, including A. salmonicida, A. hydrophila, Saprolegnia sp., Clostridium sp., Edwardsiella tarda, Photobacterium damselae, and different Vibrio species (Verschuere et al., 2000; Irianto & Austin, 2002; Kolndadacha et al., 2011; Singh, Kumari & Reddy, 2015; Amoah et al., 2019; Kuebutornye et al., 2020). Similarly, strains isolated from algae have demonstrated significant biocontrol potential in vitro. For instance, Bacillus subtilis strain IB.6a.1, isolated from Sargassum spp., inhibited methicillin-resistant S. aureus (MRSA) and Staphylococcus epidermidis, with inhibition halos of 3.75 mm and 5.3 mm, respectively (Susilowati, Sabdono & Widowati, 2015). Additionally, Prieto et al. (2012) identified Bacillus strains from red, brown, and green algae that inhibited pathogens such as MRSA, E. coli, and L. monocytogenes, producing inhibition halos exceeding three mm in radius. On the other hand, Pseudomonas fluorescens strain AH2 inhibited A. salmonicida with inhibition zones of 21 mm (Gram et al., 2001), while Pseudomonas strain NA_1, isolated from Splachnidium rugosum, inhibited pathogens such as Bacillus cereus, S. epidermidis and Pseudomonas putida with zones up to 10 mm (Albakosh et al., 2016). These results support the conclusion that the strains tested in this study have significant potential as biocontrol agents in aquaculture, particularly against pathogens such as S. aureus.

Nevertheless, to be suitable for use in aquaculture, a candidate probiotic bacterium must not only demonstrate biocontrol activity but also meet essential criteria to ensure its quality, efficacy, and safety to the host. These criteria include: (1) absence of pathogenicity or unfavorable side effects, (2) lack of drug or antibiotic resistance, (3) survival both inside and outside the host digestive tract, and (4) the ability to provide a final product containing sufficient viable probiotics to confer benefits to the host (FAO & WHO, 2006; Mujeeb et al., 2022).

The first two conditions are critical for ensuring the safety of probiotics. The bacteria must not exhibit any pathogenic factors or cause unwanted effects, such as biofilm formation, a characteristic common in pathogens like Aeromonas and Vibrio. Biofilms facilitate pathogens to colonize tissues, resist antibacterial agents, and evade host immune responses, complicating disease control (Rosini & Margarit, 2015; Cai & Arias, 2017; Graf et al., 2019; Arunkumar et al., 2020; Reichling, 2020; Cheong et al., 2021). Furthermore, biofilms on the surfaces of aquaculture systems can serve as reservoirs for pathogens, increasing disease risks for aquatic species (Cai & Arias, 2017; Freitas de Oliveira, Moreira & Schneider, 2019). Additionally, probiotic candidates must also exhibit sensitivity to antibiotics, as preventing the transfer of antibiotic resistance genes to pathogens or the gut microbiome is essential for mitigating the spread of antimicrobial resistance (Sanders et al., 2010; Mujeeb et al., 2022), particularly in aquatic environments, which serve as conduits for the dissemination of these genes across ecosystems (Cabrera-Alaix et al., 2023).

The third condition ensures that probiotics can persist in aquaculture systems and effectively colonize the target organs of the host, which are typically the digestive system (Endo & Gueimonde, 2015; Pang, Ransangan & Hatai, 2020; Diwan, Harke & Panche, 2023). Probiotic strains must survive the pH fluctuations along the gastrointestinal tract, which range from acidic in the stomach to alkaline in the intestine (Ding & Shah, 2007; Endo & Gueimonde, 2015; Solovyev et al., 2015; Pang, Ransangan & Hatai, 2020; Wendel, 2022), as well as the temperature fluctuations typical of aquatic ecosystems, which are increasingly exacerbated by climate change (Mugwanya et al., 2022). Moreover, probiotic products must also survive industrial processing. This is a significant challenge, as heat during processing can cause microbial death, thereby reducing product efficacy (Kosin & Rakshit, 2010; Sanders et al., 2010; Bidhan et al., 2014; Pang, Ransangan & Hatai, 2020; Wendel, 2022). Finally, the fourth requirement ensures that the product delivers the expected health benefits while meeting industry standards. To achieve this, it is recommended that the final product contains a minimum of 106 CFU/mL of viable probiotics (Tripathi & Giri, 2014; Jiang et al., 2018).

Based on these criteria, the biofilm formation tests revealed that none of the selected strains exhibited the capacity to form biofilms, a favorable trait for their consideration as probiotic candidates. While biofilm formation has been proposed as beneficial for probiotics by promoting intestinal colonization and prolonging their persistence in host mucosa (Salas-Jara et al., 2016), as well as supporting balanced nitrogen and carbon cycles in aquaculture systems (Cai & Arias, 2017), these structures are also strongly associated with bacterial infections in aquatic organisms and humans (Arunkumar et al., 2020; Barzegari et al., 2020; Reichling, 2020). Biofilms can detach, disperse, and adhere to other host areas, such as wounds, causing infection recurrence and economic losses (Arunkumar et al., 2020; Reichling, 2020). Additionally, biofilms act as reservoirs for pathogens, enabling them to resist disinfectants and antibiotics, thereby exacerbating disease control challenges (Cai & Arias, 2017; Freitas de Oliveira, Moreira & Schneider, 2019). Therefore, The inability of the four selected strains to form biofilms thus suggests a reduced risk of pathogenicity. However, further studies on other pathogenicity factors, such as motility, capsule formation, or hemolysin production, are necessary to confirm their safety (Pasachova-Garzón, Ramirez-Martinez & Muñoz Molina, 2019; Paz-Zarza et al., 2019; Sarkodie, Zhou & Chu, 2019). Additionally, in vivo evaluations are also essential to assess their impact on different aquaculture species.

In antibiotic susceptibility tests, most Bacillus isolates demonstrated sensitivity to all tested antibiotics, except Bacillus sp. AB17, which was resistant to streptomycin. P. mosselii AR37, however, exhibited resistance to multiple antibiotics, including streptomycin, tetracycline, oxytetracycline, ampicillin, chloramphenicol, and penicillin G (Table 2). These findings suggest that Bacillus sp. AN35 and AB08 meet the safety criterion of lacking antibiotic resistance genes, a key factor in preventing the potential spread of resistance and the emergence of new resistant pathogens (Chauhan & Singh, 2019; Uzun Yaylacı, 2022), making them promising candidates for probiotic evaluation. However, isolates resistant to a limited number of antibiotics, such as Bacillus sp. AB17, which is only resistant to streptomycin, should not be dismissed without further analysis to determine whether resistance is intrinsic or acquired through horizontal gene transfer (Endo & Gueimonde, 2015). Intrinsic resistance, encoded in the core genome of the microorganism, unlike acquired resistance obtained through horizontal gene transfer (Langendonk, Neill & Fothergill, 2021) poses minimal safety risks and a low likelihood of transfer (Compaoré et al., 2013; Endo & Gueimonde, 2015). This type of resistance has been documented in other Bacillus species (Compaoré et al., 2013), including B. subtilis SOM8, which was intrinsically resistant to streptomycin (Zhao et al., 2024), the same antibiotic to which Bacillus sp. AB17 was resistant. Confirming whether the resistance observed in Bacillus sp. AB17 is intrinsic is crucial to avoid prematurely discarding potentially safe and effective probiotic candidates.

Tolerance tests revealed that temperature and pH significantly influenced bacterial survival and growth (Figs. 5 and 6). The P. mosselii strain AR37 exhibited a narrow survival range, tolerating temperatures between 25 and 37 °C and pH levels between five and nine, with growth ranging from 107 to 109 CFU/mL. These results suggest that AR37 may not withstand the processing conditions of probiotic products or the acidic environment of the animal stomach. Consistent with these findings, Dieppois et al. (2015) and Leelagud et al. (2024) reported that members of the genus Pseudomonas, including P. entomophila, a close relative of P. mosselii, generally tolerate temperatures ranging from four to 42 °C. This aligns with the observation that AR37 was unable to survive above 45 °C. However, the pH tolerance results of this strain differ from those observed by Devi et al. (2022), who reported that P. mosselii COFCAU_PMP5 survived in the pH range of 2–9.

In the case of Bacillus sp. AB17, although it demonstrated a remarkable tolerance across a wide temperature range (25−55 °C) and pH levels (2–9), its count at pH 2 was significantly lower compared to the other pH values (p < 0.0001, 105 UFC/mL), suggesting that it may not meet the minimum threshold for probiotic efficacy in highly acidic environments such as the stomach. Meanwhile, Bacillus sp. AB08 and AN35 demonstrated robust survival under the same conditions, maintaining counts between 106 and 109 CFU/mL. This high tolerance aligns with studies highlighting the ability of many probiotic Bacillus strains to survive extreme pH and temperature conditions, supported by their ability to form endospores, which enhance survival in harsh environments (Amoah et al., 2019; Butkhot et al., 2020; Zhang et al., 2020; Shah et al., 2021). These findings position Bacillus sp. AB08 and AN35 as promising candidates for probiotic applications, as they maintain counts above the recommended minimum of 106 CFU/mL under adverse environmental conditions, an essential requirement for effective use in aquaculture (Ding & Shah, 2007; Endo & Gueimonde, 2015).

Conclusions

Mangrove seaweeds, an underexplored source of potentially probiotic bacteria, were identified as a valuable reservoir of microorganisms with biocontrol capacity, as evidenced by the finding of 12 strains (21.42% of isolates) that exhibited antibacterial activity against S. aureus and A. hydrophila. Among these, Bacillus sp. AB08 and Bacillus sp. AN35 emerged as the most promising probiotic candidates for aquaculture, displaying remarkable biocontrol activity against S. aureus, susceptibility to all tested antibiotics, inability to form biofilms, and the ability to maintain viable counts above 106 CFU/mL under challenging temperature and pH conditions.

As this study constitutes a preliminary evaluation, further research is needed to assess additional probiotic traits, such as bile tolerance and other gastrointestinal stress factors, alongside the investigation of pathogenicity factors like capsule or hemolysin production. In vivo assays are also essential to validate the efficacy of Bacillus sp. AB08 and AN35 in aquaculture systems, their impact on aquatic organisms, and their interactions with native microbiota, ensuring their safety and efficacy for future commercial applications.

Supplemental Information

Supplemental Information 1 Biocontrol testing against fish and shellfish pathogens

Supplemental Information 2 Biofilm formation test

Supplemental Information 3 Antibiotic susceptibility test

Supplemental Information 4 Tolerance testing for temperature

Supplemental Information 5 Tolerance testing for pH

Supplemental Information 6 Sequences of algae-associated bacteria identified

We would like to express our gratitude to the staff of the Microbiological Research Laboratory (LIM) and the Plant and Microorganism Biology research group for their invaluable collaboration. The AI tool DeepL was used for text editing.

Additional Information and Declarations

Competing Interests

Author Contributions

Field Study Permissions

DNA Deposition

Data Availability

The authors declare there are no competing interests.

Juliana Martinez-Delgado conceived and designed the experiments, performed the experiments, analyzed the data, prepared figures and/or tables, authored or reviewed drafts of the article, and approved the final draft.

Neyla Benitez-Campo conceived and designed the experiments, analyzed the data, authored or reviewed drafts of the article, and approved the final draft.

The following information was supplied relating to field study approvals (i.e., approving body and any reference numbers):

Field experiments were approved by Autoridad Nacional de Licencias Ambientales (ANLA).

The following information was supplied regarding the deposition of DNA sequences:

The sequences of the algal-associated bacteria are available in the Supplementary File and at GenBank: PQ276436 to PQ276487.

The following information was supplied regarding data availability:

The raw data for biocontrol tests, biofilm formation tests, antibiotic susceptibility tests and temperature and pH tolerance tests, and the sequences of the identified bacteria are available in the Supplementary Files.

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
