# Peer review of "Probiotic potential of bacteria associated with the mangrove epiphytic algae Bostrychia calliptera and Rhizoclonium riparium"

_PeerJ, doi:10.7717/peerj.19073_

## Round 0.1 · original submission · Major Revisions

the paper focuses on a topical issue but needs to be rewritten more academically. In the material and methods, the ethics committee's approval concerning fish studies has to be added. In general, the text can be condensed. In addition, the support of scientific references is often missing. I recommend authors re-read and apply the journal's guidelines to the revised manuscript.

Reviewer 1 ·

Basic reporting

Figure 6 in line 386 is about temperature tolerance data, while this section requires pH tolerance data. Please check the relevant description and corresponding chart numbers

Experimental design

Why didn't the author directly use a screening plate to screen for bacteria with antibacterial effects after enrichment culture in the initial screening stage, instead of using colonies picked from ordinary plates for antibacterial experiments?

Validity of the findings

Fig3 shows the inhibition zones of some bacteria, but there is no specific display of the inhibition zones of the two well performing strains of Pseudomonas and Bacillus selected in this article. Please add at least the inhibition images of the final screened target strains in the revised manuscript.

Additional comments

1. The author isolated microorganisms from mangrove species rich in microorganisms, but only obtained 52 strains of bacteria belonging to 9 species, which is a small number. Is it because the author overlooked some bacteria when selecting representative colonies for identification?
2.Line 358 is an analysis of the ability of bacteria to form biofilms, but there is only one conclusion in this section that does not form biofilms. The rest is data on antibiotic sensitivity experiments. Please revise the wording and title of this section
3. The discussion section contains a large number of results describing the distribution of algae related bacteria. Please focus on comparing with the results of this question, analyzing the phenomena and causes.

Reviewer 2 ·

Basic reporting

The English usage needs revision. The flow of the article is missing in the introduction. The overall comment is that the manuscript can be accepted if major revisions are made

Experimental design

The research question is not defined. Needs revision of methods. See how materials and methods are written in journals

Validity of the findings

Check the language and mode of writing of the manuscript. The result section can be improved

Annotated reviews are not available for download in order to protect the identity of reviewers who chose to remain anonymous.

---

## Round 0.2 · accepted · Accept

The manuscript and the English have been thoroughly revised and improved, and all the changes suggested by the reviewers have been taken into account.

I am pleased to confirm that your paper has been accepted for publication in PeerJ.

Reviewer 1 ·

Basic reporting

No

Experimental design

No

Validity of the findings

No

Additional comments

No